# Effects of 1 MHz Therapeutic Ultrasound on Limb Blood Flow and Microvascular Reactivity: A Randomized Pilot Trial

**DOI:** 10.3390/ijerph182111444

**Published:** 2021-10-30

**Authors:** Megan Waters, Branko Miljkovic, Jozelyn Rascon, Manuel Gomez, Alvaro N. Gurovich

**Affiliations:** 1Doctor of Physical Therapy Program, College of Health Sciences, The University of Texas at El Paso, El Paso, TX 79968, USA; mwaters@miners.utep.edu (M.W.); bmiljkovic@miners.utep.edu (B.M.); jrascon3@miners.utep.edu (J.R.); 2Clinical Applied Physiology Laboratory, College of Health Sciences, The University of Texas at El Paso, El Paso, TX 79902, USA; mgomez26@miners.utep.edu

**Keywords:** therapeutic ultrasound, blood flow, microvascular function

## Abstract

A randomized, double-blind, placebo-controlled, cross-over study where continuous therapeutic ultrasound (CUS; at 0.4 W/cm^2^), pulsed therapeutic ultrasound (PUS; at 20% duty cycle, 0.08 W/cm^2^), both at 1 MHz, and placebo (equipment on, no energy provided) were randomized and applied over the forearm of the non-dominant arm for 5 min in 10 young, healthy individuals. Absolute and peak forearm blood flow (FBF) were measured via Venous Occlusion Plethysmography. FBF was measured before, halfway, and after (immediately and 5 min after) the therapeutic ultrasound (TUS) intervention. Post-ischemic peak FBF was measured 10 min before and 10 min after the TUS intervention. A two-way repeated measures ANOVA (group × time) was selected to assess differences in FBF before, during, and after TUS treatment, and for peak FBF before and after TUS treatment. FBF increased 5 min after TUS in CUS compared to placebo (2.96 ± 1.04 vs. 2.09 ± 0.63 mL/min/100 mL of tissue, *p* < 0.05). PUS resulted in the greatest increase in Peak FBF at 10 min after US (Δ = 3.96 ± 2.02 mL/min/100 mL of tissue, *p* = 0.06). CUS at 1 MHz was an effective treatment modality for increasing FBF up to 5 min after intervention, but PUS resulted in the greatest increase in peak FBF at 10 min after intervention.

## 1. Introduction

Musculoskeletal injuries are typically associated with tissue damage, pain, and inflammation [1]. Resolution of soft tissue damage relies on endothelium-dependent blood flow to deliver essential cellular components and healing factors [2]. Vascular endothelial cells play a key role in regulating local blood flow and maintaining vascular integrity through the production of nitric oxide (NO) in response to physical or chemical stimuli [3,4]. NO regulates vascular tone and permeability, inhibits platelet aggregation, and stimulates angiogenesis [4,5,6,7]. NO synthesis during tissue healing has a positive clinical implication as it promotes aforementioned essential components required for maintaining hematological stasis around the injured tissue [1].

Physical agents, or modalities, are usually incorporated into the plan of care for treating soft tissue injuries in rehabilitation with the goal of mitigating pain and inflammation by increasing blood flow [8,9]. For more than 50 years, therapeutic ultrasound (TUS) has been one of the most used modalities in rehabilitation [10,11,12,13,14]. TUS is a mechanical sound wave that is transmitted through the tissues by absorption, reflection, and refraction [8,10,11]. These mechanical forces produce a combination of cavitation and acoustic microstreaming effects, which elicit stimulation of fibroblastic activity and collagen synthesis, increased blood flow, and enhanced tissue regeneration [9,15,16,17]. However, different low-frequency TUS waves may generate different outcomes, especially when measuring blood flow. The increase in blood flow produced by TUS is normally attributed to an increase in internal temperature [15,17,18]. However, non-thermal TUS has also been shown to increase blood flow, which may be related to NO production [3,16,19]. In summary, the effects of TUS in endothelial-dependent blood flow are yet to be determined.

An adequate understanding of the effects of TUS on both blood flow and endothelial function would help rehabilitation professionals to determine the best plan of care when treating musculoskeletal injuries. To our best knowledge, there have been no studies investigating the effects of TUS on blood flow in the microcirculation (e.g., muscle capillaries). Therefore, the primary objective of this study is to compare the effects of continuous and pulsed TUS waveforms on changes in blood flow at the microvascular level. Our second objective is to determine if changes in blood flow are endothelial-dependent, which is associated with NO bioavailability.

## 2. Materials and Methods

### 2.1. Participant Characteristics and Inclusion Criteria

Participant recruitment began in October 2019, and the trial was conducted from January to March 2020. The power analysis estimated the sample size using G*Power 3.1 [20,21] with an effect size of 0.75 (75% of change or difference between groups), a power of 0.80, and a significance level of 0.05 (two sided) (n = 8/group). Healthy volunteers between the ages of 18 and 35, and of any racial or ethnic background, were recruited from The University of Texas at El Paso and its surroundings. Inclusion criteria required participants to be non-smokers, have a body mass index (BMI) of <30 kg/m^2^, and a resting blood pressure not exceeding 140/90 mm Hg. Exclusion criteria included any injury to the non-dominant upper extremity within the last 6 months, any cutaneous/subcutaneous lesion to the non-dominant upper extremity, any known cardiovascular disease such as coronary artery disease or peripheral vascular disease, any form of known skeletal muscle, rheumatic, metabolic, neurological, immunological, hematological, or psychiatric disorders, and any prescription medication medicines (except for birth control medicines). If subjects were taking over-the-counter painkillers, such as NSAIDs or aspirin, or nutritional supplements containing antioxidants, we asked them to abstain from their use for 12 h prior to each visit. Inclusion and exclusion criteria were determined using a preliminary screening form prior to participation. The study was approved by the Institutional Review Board of The University of Texas at El Paso, and all individual participants included in the study signed the informed consent before scheduling the first visit to the Clinical Applied Physiology (CAPh) Lab.

Women recruited for this study were tested during the early follicular (0–4 days after onset of menses) or late luteal (24–28 days after onset of menses) phases of their menstrual cycle to eliminate any influence these two phases had on blood pressure and blood flow [22,23]. The Preliminary Screening form allowed to determine their hormonal cycle regularity and the beginning of the cycle with ±4 days of accuracy. Based on the aforementioned information, we scheduled the participants’ visits to coincide with the late luteal phase.

### 2.2. Experimental Design

A randomized, double-blinded, placebo-controlled, cross-over, repeated-measures design was conducted to measure blood flow changes in the non-dominant forearm before (baseline), during, and after each intervention. This study required 3 1-h visits to the CAPh Lab. Interventions (i.e., placebo, continuous TUS, and pulsed TUS) were randomly assigned and applied to the participants. Randomization was achieved using a random number generation online application (https://numbergenerator.org/randomnumbergenerator/1-3, accessed on 29 October 2021). Each participant served as his/her own control. The participant, TUS technician, and vascular testing technician were blinded to treatment order, and the TUS technician was blinded to all vascular tests obtained before, during, and after treatments.

### 2.3. Experimental Procedures

Height (seca, Hamburg, Germany) and weight (WB-110A Class III, Tanita, Japan) to determine BMI, as well as resting blood pressure (BP760, Omron Healthcare, Inc., Lake Forest, IL, USA) were assessed at baseline in the first lab visit. If the subject did not meet the inclusion criteria for BMI and blood pressure, she/he was withdrawn from the study. Resting blood pressure was measured at the start of each lab visit to ensure continued eligibility. Resting blood pressure was determined by taking the average of 3 measurements after a 10 min resting period.

All participants underwent 3 testing sessions on separate days with 48 h between sessions. Once the subject was recruited and signed the informed consent, preparations for the vascular assessment during resting conditions began. All three treatment sessions followed the same procedures. Venous occlusion plethysmography (VOP) (AI6 Arterial Inflow System, D. E. Hokanson, Inc., Bellevue, WA, USA), the gold standard for microvascular in vivo blood flow assessment [24], was performed on the participant’s non-dominant upper extremity using a calibrated silicone-type band (i.e., mercury strain gauges) and two pressure cuffs placed on the upper arm and on the wrist (Figure 1). The strain gauge was applied to the non-dominant forearm at the larger forearm circumference. The participant was instructed to lie supine with the upper extremity passively elevated above heart level. Absolute forearm blood flow (FBF) was measured before, during (half-way at 2.5 min), and after (immediately and 5 min after) the 5 min TUS intervention. Peak FBF (or peak vascular reactivity) was measured 10 min before and 10 min after the TUS intervention.

VOP protocol has been described in previous studies [22,25]. The participant remained lying briefly in the same position for 20 min in order to obtain stable baseline measurements of FBF. The upper arm cuff inflation cycled from 0 to 50 mm Hg for 7 s every 15 s to prevent venous outflow. One minute before each measurement, the wrist cuff was inflated to 250 mm Hg to occlude hand circulation. Absolute FBF was determined by the rate of change of limb circumference (e.g., slope) during the seven-second venous occlusion. FBF was estimated as the average of 4 readings in 1 min [22,26,27,28].

Peak FBF during reactive hyperemia was measured after a 5 min blood flow occlusion. These measurements are a reliable non-invasive alternative to estimate endothelial function in resistance vessels, a biomarker for NO bioavailability [27]. Baseline FBF was recorded for 2 min, then the cuff on the upper arm was inflated to 200 mm Hg for 5 min to induce ischemia and then rapidly deflated after 5 min to produce reactive hyperemia. Following deflation, FBF was measured every 15 s for three minutes; peak FBF was selected from the highest FBF following deflation of the cuff, using the same slope analysis as described above.

TUS (Chattanooga Intelect Legend XT, DJO, LLC, Lewisville, TX, USA) treatments (placebo, pulsed, or continuous) were initiated once baseline FBF measurements were taken and determined to be stable. Continuous waveforms were applied at a spatial average temporal intensity (SATA) of 0.4 W/cm^2^ using a 5 cm^2^ transducer. Pulsed waveforms were applied with a 20% duty cycle (i.e., 2 ms on, 8 ms off), representing a SATA intensity of 0.08 W/cm^2^. Placebo TUS had the unit on, but no US energy was provided. The TUS transducer was applied for 5 min at the same point over the forearm, above the strain gauge, and was moved with synchronic movements at ~4cm/s [8]. Commercially available ultrasound gel was used as a conduction agent.

### 2.4. Statistical Analysis

Descriptive statistics, including mean and standard deviations, were obtained. Normal distribution for all dependent variables was evaluated by Kolmogorov-Smirnov test. The assumption of sphericity was checked by Mauchly’s test and corrected by Greenhouse-Geisser method when it was necessary. A two-way repeated measures ANOVA (group × time) was selected to assess differences in FBF before, during, and after TUS treatment and for peak FBF before and after TUS treatment. The statistical analysis was performed with SPSS (version 25.0, IBM, Chicago, IL, USA) and significance was set at *p* < 0.05.

## 3. Results

In this case, 10 participants (six males, four females) were randomly assigned, received intended treatments, and included in the analysis. Data were normally distributed, and the assumption of sphericity was confirmed. Data are presented as averages and standard deviations, unless specifically noted. Demographics and general characteristics of the sample are shown in Table 1.

FBF increased 5 min after TUS in CUS compared to placebo and PUS (2.96 ± 1.04 vs. 2.09 ± 0.63 mL/min/100 mL of tissue and 2.31 ± 0.62 mL/min/100 mL of tissue, *p* = 0.021 and *p* = 0.078, respectively). There were no significant differences in FBF with any of the TUS treatments halfway through or immediately after TUS intervention (Figure 2).

PUS resulted in the greatest increase in peak FBF at 10 min after TUS (Δ = 3.96 ± 2.02 mL/min/100 mL of tissue, *p* = 0.060). Placebo and CUS displayed no increase in peak FBF at 10 min after (Figure 3).

No adverse effects were reported by the volunteers or observed by the researchers during the length of the study.

## 4. Discussion

The purpose of the current study was to compare the effects of continuous and pulsed TUS waveforms on changes in blood flow at the microvascular level, as well as to determine if any changes in blood flow are endothelial-dependent and associated with NO bioavailability. To the best of our knowledge, this is the first study investigating the effects of therapeutic ultrasound on microvascular circulation. The results found that, in healthy volunteers, CUS might increase microvascular blood flow after the TUS application and that PUS increased peak vascular reactivity, which is associated with an increase in endothelial function and NO bioavailability.

The increase in blood flow produced by TUS is normally attributed to an increase in internal temperature due to the absorption of ultrasonic waves [15,17,18]. However, blood flow has also been shown to increase as a result of non-thermal effects associated with the mechanical forces producing the combination of cavitation and acoustic microstreaming effects, which may be related to NO production [3,16,19]. The thermal and mechanical effects occur simultaneously; however, the effects are dose-dependent on application parameters. Results of extensive in vitro studies suggest that therapeutic TUS interacts with one or more of the inflammatory components, potentially resulting in earlier resolution of inflammation [15,18]. Possible mechanisms include stimulation of macrophage-derived factors, heightened fibroblast recruitment, accelerated fibrinolysis, accelerated angiogenesis, and increased collagen and matrix synthesis [9,15,17,18,29,30]. Even though the present study showed a decrease in FBF during the intervention (Figure 2), CUS produced a re-bounce on blood flow 5 min after the intervention. This finding could be explained by two reasons: (1) as the upper extremity during VOP needs to be elevated during the assessment of blood flow, subjects kept their forearm in that position for the total time of the study; therefore, there might be some decrease in blood flow due to gravity and (2) the re-bounce of blood flow is a clear effect of CUS enhancing blood flow even with gravity affecting it. On the other hand, vascular reactivity increased after PUS confirming that non-thermal TUS might be associated with endothelial function and NO bioavailability, rather than thermal TUS.

Experimental studies on humans have demonstrated that continuous and pulsed TUS waveforms at 1 MHz and 3 MHz improved endothelium-dependent vasodilation of the brachial artery [3,19]. In a similar study, Cruz et al. (2016) investigated the effects of therapeutic ultrasound of 1 MHz at the same SATA intensity of 0.4 W/cm^2^ for both CUS and PUS (20% duty cycle) waveforms on endothelial function at the macrovascular level by measuring brachial flow-mediated dilation (FMD) [3]. Compared to the placebo, FMD was increased by 4.8% using continuous ultrasound (*p* < 0.001) and by 3.4% using pulsed ultrasound (*p* < 0.001) [3]. The study showed no change in skin temperature (group: *p* = 0.837; time *p* = 0.190; interaction: *p* = 0.788) after ultrasound application for both CUS and PUS, suggesting the increase in vasodilation to be a result of mechanical effects and not thermal [3]. The study concluded that both CUS and PUS waveforms cause endothelium-dependent arterial vasodilation by increased production of NO.

Similarly, Hauck et al. [19] compared the effects of 1 MHz and 3 MHz waveforms using both CUS (SATA 0.4 W/cm^2^) and PUS (20% duty cycle) in 2 groups of 15 subjects each. Both 1 MHz (CUS: Δ4%, *p* < 0.001; PUS: Δ3%, *p* < 0.001) and 3 MHz (CUS: Δ4%, *p* < 0.001; PUS: Δ3%, *p* < 0.001) TUS waveforms increased FMD by ~4% compared to placebo [19]. They found no differences in secondary outcomes for baseline diameter, hyperemic flow, or nitroglycerin-mediated diameter and vasodilation [19]. The results found by Hauck et al. [19] are in agreement with Cruz et al. [3] and further suggest that the vasodilation was due to the mechanical effects of the TUS, specifically by production of NO [3,19]. These findings support the use of TUS in promoting tissue repair through improved macrovascular function; however, improved macrovascular function does not infer improved tissue blood flow or enhanced microcirculation. Based on our best knowledge, there is only one study investigating therapeutic agents and microcirculation [31]. Chang et al. [31] studied four therapeutic physical agents (i.e., TUS, laser, interferential current, and vibration massage) on the Achilles tendon microcirculation. This cross-sectional study included 51 young and healthy individuals that were randomly assigned to one of four intervention groups. Their results showed that only TUS and vibration massage increased Achilles tendon microcirculation, confirming the benefits of mechanical stimulation over electrical currents and laser [31]. The results of the present study are similar to those from Chang et al. [31]; however, they assessed microcirculation via near infrared spectroscopy (NIRS) in a more limited area than the current study, where the current study used VOP in a broader volume. These differences could explain the more robust results from Chang et al. [31] when compared with the current results.

Modulated microcirculation has been shown to accelerate healing by controlling ischemia, hypoxia, edema, and local secondary tissue damage [1,2]. There have been a few studies investigating the effects of physical agents in larger microcirculation beds; however, these studies were laser therapy-induced, not TUS-induced [25,32]. For example, Larkin et al. [25] studied the dose response effect of laser therapy on forearm blood flow using venous occlusion plethysmography (VOP) to measure changes in limb circulation in 10 young, healthy men. Their main results showed an increase in tissue blood flow during treatment with 360-J when compared to sham, 180-J, and 720-J, but the effects went back to baseline values 3 min after treatment [25]. Additionally, Maegawa et al. [32], using an animal model of the mesenteric circulation, found that low-level laser irradiation (~300-J at 830 nm) increased microvascular blood flow and, at least in the first 5 min, it was NO dependent [32].

The present study provides insight into implementing specific TUS parameters to increase limb blood flow in vivo. CUS elicited an increase in FBF for at least 5 min after treatment, longer than laser therapy [25]. Moreover, PUS improved endothelial function of the microvasculature for at least 10 min after treatment cessation. These findings suggest that the mechanical effects of therapeutic TUS might induce shear stress over endothelial cells, enhancing NO bioavailability to a larger extent compared to the photobiomodulation effects of laser therapy [19,25]. Based on the findings of the present study and previous reports [15,17,19], continuous and pulsed 1 MHz therapeutic ultrasound could elicit microvascular effects by mechanically stimulating endothelial cells to produce NO [15,17,19]. NO synthesis during tissue healing regulates vascular tone and permeability, inhibits platelet aggregation, and stimulates angiogenesis resulting in hematological stasis around the injured tissue [1,4,5,6,7]. Furthermore, the results of the present study might provide some justification for the treatment of soft tissue injuries utilizing continuous or pulsed 1 MHz ultrasound parameters when increasing microvascular blood flow is the main purpose.

### Limitations and Future Studies

Even though the study was designed to test at least 16 subjects (8 males and 8 females), the sample size was limited to only 10 participants due to COVID-19 halting human research in the Spring of 2020. The lack of deep tissue temperature measurements prevents us from ruling out the possibility of thermal effects associated with CUS. In addition, we measured absolute FBF only up to 5 min after TUS intervention to be able to test endothelial function after intervention. Therefore, the extent to which blood flow may have continued increasing with CUS remains unknown. Moreover, the elevation of the forearm during the whole intervention might have had an impact on blood flow; therefore, future studies should consider performing the intervention on a flat surface and elevate the forearm only during VOP measurements. Finally, we tested only healthy subjects. Future studies should measure deep tissue temperature and changes in blood flow for prolonged periods of time. Such outcomes would provide greater insight for determining appropriate TUS dosing. It would then be beneficial for future authors to investigate the effects of TUS in clinical populations with musculoskeletal conditions. Even though COVID-19 restrictions limited our sample size, the results of the present study allowed us to use this data as a pilot study and restructure some of the study design.

## 5. Conclusions

In conclusion, the results of the present pilot study showed that TUS waveform and wave frequency parameters might result in an increased microvascular blood flow and an increased peak vascular reactivity in healthy adults. Therefore, the findings indicate that 1 MHz continuous and pulsed waveforms might be effective for increasing FBF and increasing peak FBF, respectively.

## Figures and Tables

**Figure 1 ijerph-18-11444-f001:**
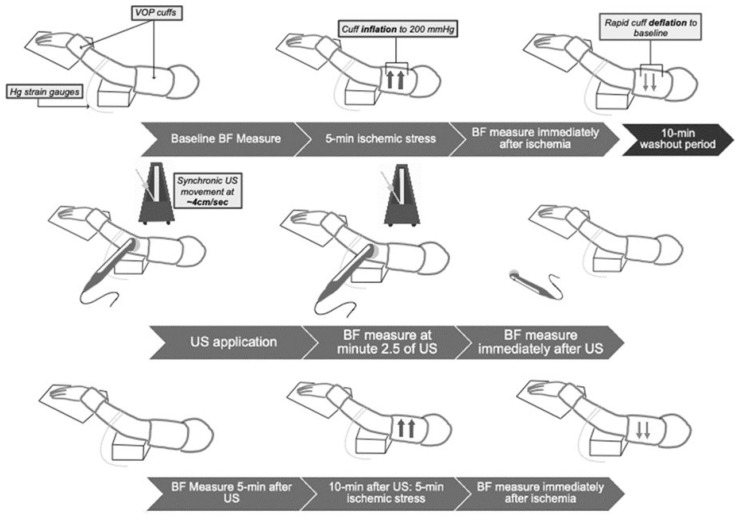
Experimental protocol. BF: blood flow; US: ultrasound; VOP: venous occlusion plethysmography.

**Figure 2 ijerph-18-11444-f002:**
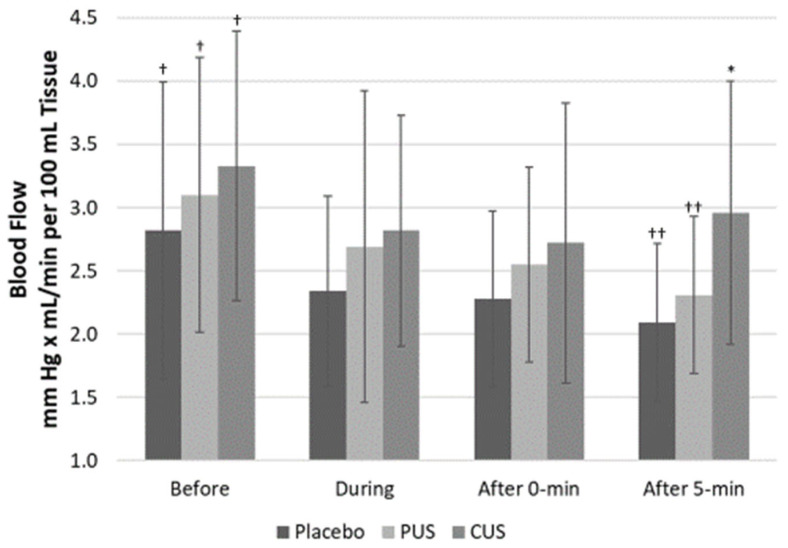
Forearm blood flow before, during, and after therapeutic ultrasound application. Data presented as average and S.D. PUS: pulsed ultrasound; CUS: continuous ultrasound; * *p* < 0.05 CUS vs. Placebo 5 min after application; †: *p* < 0.05 Placebo, PUS, and CUS before vs. 0 min after application; ††: *p* < 0.05 Placebo and PUS before vs. 5 min after application.

**Figure 3 ijerph-18-11444-f003:**
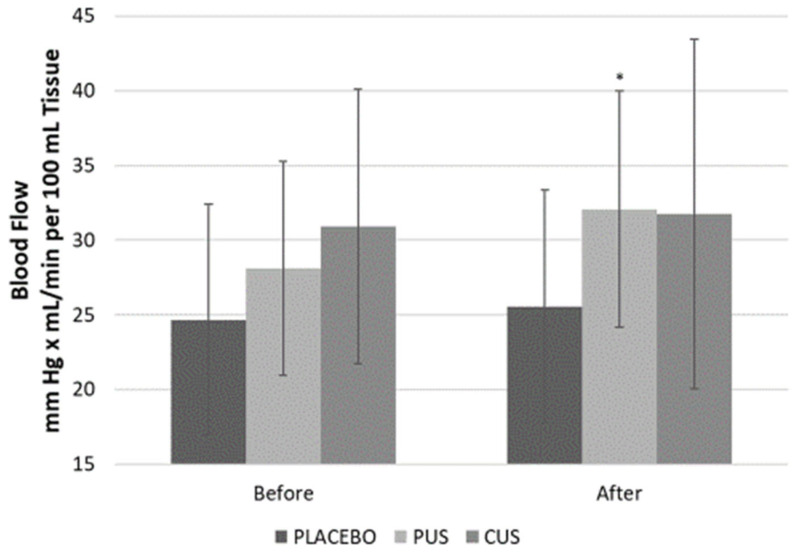
Peak forearm blood flow after reactive hyperemia before and after therapeutic ultrasound application. Data presented as average and S.D. PUS: pulsed ultrasound; CUS: continuous ultrasound; * *p* < 0.05 before vs. after in PUS.

**Table 1 ijerph-18-11444-t001:** Sample characteristics.

Demographics	Mean	Std. Deviation
Age (years)	25.90	3.90
Weight (kg)	72.25	10.37
Height (cm)	163.87	8.27
Body Mass Index (kg/m^2^)	26.83	2.54
Systolic BP (mm Hg)	122.97	11.88
Diastolic BP (mm Hg)	76.5	9.24

## Data Availability

Raw data from this study can be found in the following link: https://minersutep-my.sharepoint.com/:b:/g/personal/agurovich_utep_edu/EWDX-NC3b-RKndlQpuP9LXEBwITZxEhiYfR6_Llj03z7fA?e=jGnRO4 (accessed on 29 October 2021).

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
