# Peer review of "Effects of 1 MHz Therapeutic Ultrasound on Limb Blood Flow and Microvascular Reactivity: A Randomized Pilot Trial"

_ijerph, 2021, doi:10.3390/ijerph182111444_

Round 1

Reviewer 1 Report

The following is the summary of the present study:

A randomized, double-blind, placebo-controlled, cross-over study where continuous (CUS; at 0.4 W/cm2), pulsed (PUS; at 20% duty cycle, 0.08 W/cm2), and placebo (equipment on, no energy provided) ultrasound at 1 MHz were randomized and applied over the forearm of the non-dominant arm for 5 minutes in 10 young, healthy individuals. A two-way repeated measures ANOVA (group x time) was selected to assess differences in FBF before, during, and after US treatment, and for peak FBF before and after US treatment. FBF increased 5 minutes after US in CUS compared to placebo (2.96 ± 1.04 vs. 2.09 ± 0.63 mL/min/100 mL of tissue, p<0.05). PUS resulted in the greatest increase in Peak FBF at 10 minutes after US (Δ=3.96 ± 2.02 mL/min/100 mL of tissue, p=0.06). CUS at 1 MHz was an effective treatment modality for increasing FBF up to 5 minutes after intervention, but PUS resulted in the greatest increase in peak FBF at 10 minutes after intervention.

The article is interesting. I have several comments:

In line 49, please simply explain or define what microcirculation is.

In line 57, please provide the IRB number of this study. By the way, as a clinical trial, is it registered in any database?

In line 61, what did the authors require the participants to be at a certain age range?

In line 77, is there a specific ratio between the number of male and that of female participants?

In line 82, please describe how to generate the random sequence.

Author Response

The article is interesting.

Thank you for your kind comments

In line 49, please simply explain or define what microcirculation is.

Thank you for your comment. We have added a brief explanation in parenthesis “(e.g. muscle capillaries)” to help defining microcirculation.

In line 57, please provide the IRB number of this study. By the way, as a clinical trial, is it registered in any database?

According to the NIH, a clinical trial must answer yes to all of these 4 questions:

  1. Does the study involve human participants?
  2. Are the participants prospectively assigned to an intervention?
  3. Is the study designed to evaluate the effect of the intervention on the participants?
  4. Is the effect being evaluated a health-related biomedical or behavioral outcome?

Our study answered yes in the first 3 but not the last one; therefore, it is not considered a clinical trial.

In line 61, what did the authors require the participants to be at a certain age range?

As stated in line 61, the age range was 18 to 35 years. The main reason is that after age 35, the vasculature might have some changes related to normal aging, which could confound the data by adding an extra variable (i.e. age). Between 18-35 years, with a health questionnaire to assess cardiovascular risk factors, we can be more confident of less variability do to aging.

In line 77, is there a specific ratio between the number of male and that of female participants?

Thank you for your comment. We wanted to recruit an even number of males and females (10 subjects per group); however, the COVID-19 pandemic halted all lab activities. That is why we are presenting this data as a pilot study. We have edited the limitation section (page 8, just before 5. Conclusions), to better address this issue.

In line 82, please describe how to generate the random sequence.

Thank you for your comment. In lines 91-101, we explain how the random sequence was generated using an online tool.

Reviewer 2 Report

The manuscript by Waters and colleagues studies the effect of therapeutic ultrasound on forearm blood flow and microvascular circulation. The authors found that continuous ultrasound increased microvascular blood flow and that pulsed ultrasound increased peak vascular reactivity, which is associated with endothelial function and NO bioavailability, after applying ultrasound in healthy volunteers. This is an interesting pilot study. However, the novelty is not given as many studies have explored that therapeutic ultrasound increases blood flow. In addition, too small sample size could be a major concern.

Author Response

We would like to thank the reviewer’s comments and kind works about our work.

The novelty is not given as many studies have explored that therapeutic ultrasound increases blood flow.

Thank you for your comment. As established in lines 50-53 on the introduction and lines 212-215 on the discussion, all studies in ultrasound and blood flow have been done in larger arteries and not in the microvasculature. That is the novelty of this study.

Too small sample size could be a major concern

We completely agree with the comment. However, our power analysis suggested that an n of 8 would be enough power and we have 10 subjects. In addition, and as addressed on the limitation section, we are presenting this data as a pilot study.

Reviewer 3 Report

  1. It's hard to read with the US acronym used so often, when people's minds jump to United States immediately after seeing/reading it. Why not use TU for Therapeutic Ultrasound?
  2. TU has been used since 1950s and yet you only cite [8,9] when referencing it. Cite more TU related publications. Take the reader on a journey to understanding TU and where it came from, before you let us read your methods.
  3. Why using dash in [8-9] and [10, 12-13], when it's just two articles apart. It's supposed to be [8, 9] and [10, 12, 13].
  4. Your experimental protocol and statistical analysis are well designed. Congratulations on that.
  5. The BMI of the participants has a low standard deviation. It would be interesting to see the effect of your procedure on different groups with different BMI categories. At least talk about this in your discussion section.
  6. There are good BMI alternatives, e.g., RFM, BAI, WC, WHR. Perhaps you should use some of those to find a group of people for whom your procedure is more or less beneficial than others. 
  7. Your graphics are blurry/pixelated. Improve the quality of your graphics. It needlessly lowers the quality of the entire paper. 

Author Response

It's hard to read with the US acronym used so often, when people's minds jump to United States immediately after seeing/reading it. Why not use TU for Therapeutic Ultrasound?

Excellent comment. Thank you very much. I completely agree with the similarity with the USA. We have replaced US with TUS along the manuscript.

TU has been used since 1950s and yet you only cite [8,9] when referencing it. Cite more TU related publications. Take the reader on a journey to understanding TU and where it came from, before you let us read your methods.

We appreciate the reviewer’s comment. We have edited the introduction with a statement reflecting the more than 50 years of TUS use and we have added relevant references to support it.

Why using dash in [8-9] and [10, 12-13], when it's just two articles apart. It's supposed to be [8, 9] and [10, 12, 13].

Thank you for your comment. This is a format issue that is controlled by the Journal. We used the recommended software for reference management and the editorial office would change it if needed.

Your experimental protocol and statistical analysis are well designed. Congratulations on that.

Thank you very much for your kind comment.

The BMI of the participants has a low standard deviation. It would be interesting to see the effect of your procedure on different groups with different BMI categories. At least talk about this in your discussion section. There are good BMI alternatives, e.g., RFM, BAI, WC, WHR. Perhaps you should use some of those to find a group of people for whom your procedure is more or less beneficial than others.

We appreciate and agree with your comment. It would be very interesting to see if different anatomical or other demographic variables can affect TUS effects. Unfortunately, the presented sample does not allow for such analysis, as the sample size would not be enough to give statistical power. We will though consider these other variables in future studies.

Your graphics are blurry/pixelated. Improve the quality of your graphics. It needlessly lowers the quality of the entire paper

We appreciate your comment. We will happy to work with the editorial office to improve the quality of the figures, once the manuscript is accepted.

Round 2

Reviewer 2 Report

Author's Notes

We would like to thank the reviewer’s comments and kind works about our work.

The novelty is not given as many studies have explored that therapeutic ultrasound increases blood flow.

Thank you for your comment. As established in lines 50-53 on the introduction and lines 212-215 on the discussion, all studies in ultrasound and blood flow have been done in larger arteries and not in the microvasculature. That is the novelty of this study.

  • In Lines 233-235, the author has mentioned that this study is not the first study on microvascular circulation.

Too small sample size could be a major concern

We completely agree with the comment. However, our power analysis suggested that an n of 8 would be enough power and we have 10 subjects. In addition, and as addressed on the limitation section, we are presenting this data as a pilot study.

  • Considering that this is a pilot study and collection of data in a difficult period, I agree to publish in the present form.